# Disparities in Body Color Adaptability and Ambient Light Color Preference between Wild and Hatchery-Reared Marbled Rockfish (*Sebastiscus marmoratus*)

**DOI:** 10.3390/ani14111701

**Published:** 2024-06-05

**Authors:** Yulu Qi, Chenhui Liu, Guozi Yuan, Haoyu Guo, Joacim Näslund, Yucheng Wang, Jiangfeng Ru, Yingying Ou, Xuejun Chai, Xiumei Zhang

**Affiliations:** 1Fisheries College, Zhejiang Ocean University, No. 1, Haida South Road, Lincheng Changzhi Island, Zhoushan 316022, China; yuluqi990913@163.com (Y.Q.); liuchenhui@stu.ouc.edu.cn (C.L.); yuanguozi@stu.ouc.edu.cn (G.Y.); 189669256309@zjou.edu.cn (Y.W.); jiangfengru@zjou.cn (J.R.); ouyingying@zjou.edu.cn (Y.O.); xiumei1227@163.com (X.Z.); 2Department of Aquatic Resources, Institute of Freshwater Research, Swedish University of Agricultural Sciences, 178 93 Drottningholm, Sweden; joacim.naslund@gmail.com; 3Key Laboratory of Mariculture & Enhancement, Marine Fisheries Research Institute of Zhejiang Province, Zhoushan 316022, China; chaixj6530@sohu.com

**Keywords:** *Sebastiscus marmoratus*, HSB model, body color variation, light environment, selective preferences

## Abstract

**Simple Summary:**

Our findings highlight significant distinctions in body coloration between wild and hatchery-reared individuals, and a ten-day rearing period under colored ambient light can notably change the coloration of hatchery-reared marbled rockfish. Additionally, our research reveals a preference among juvenile marbled rockfish for a red-light environment, alongside a consistent negative phototactic response to yellow and blue light colors. These results emphasize the substantial influence of hatchery rearing conditions on fish body color and morphological color-changing abilities, and elucidate ambient light color preferences. Our study emphasizes the importance of considering modifications to the hatchery environment, particularly in regulating ambient light color, within stock enhancement programs.

**Abstract:**

Hatchery rearing significantly influences the phenotypic development of fish, with potential adverse effects for the post-release performance of hatchery-reared individuals in natural environments, especially when targeted for stock enhancement. To assess the suitability of releasing hatchery-reared fish, a comprehensive understanding of the phenotypic effects of captive rearing, through comparisons with their wild conspecifics, is essential. In this study, we investigated the divergence in body coloration between wild and hatchery-reared marbled rockfish *Sebastiscus marmoratus*. We examined the selection preferences for different light colors and assessed the impact of different ambient light colors on the morphological color-changing ability of juvenile marbled rockfish. Our findings revealed significant differences in body color between wild and hatchery-reared marbled rockfish. The hue and saturation values of wild marbled rockfish were significantly higher than those of their hatchery-reared counterparts, indicative of deeper and more vibrant body coloration in the wild population. Following a ten-day rearing period under various light color environments, the color of wild marbled rockfish remained relatively unchanged. In contrast, hatchery-reared marbled rockfish tended to change their color, albeit not reaching wild-like coloration. Light color preference tests demonstrated that wild juvenile marbled rockfish exhibited a preference for a red-light environment, while hatchery-reared individuals showed a similar but weaker response. Both wild and hatchery-reared marbled rockfish displayed notable negative phototaxis in the presence of yellow and blue ambient light. These results highlight the impact of hatchery rearing conditions on the body color and morphological color-changing ability, and provide insight into light color selection preferences of marbled rockfish. To mitigate the divergence in phenotypic development and produce more wild-like fish for stocking purposes, modifications to the hatchery environment, such as the regulation of ambient light color, should be considered.

## 1. Introduction

Hatchery-reared fish may develop maladaptive changes in biological traits, through plasticity or selection, potentially hampering their survival ability in the wild following release [1,2]. These traits encompass a range of characteristics, including morphological, physiological, and behavioral traits, differing between hatchery-reared individuals and their wild counterparts [3,4,5]. Among the identifiable morphological divergences observed in cultured fish, alterations in body coloration stand out as particularly noticeable, common, and potentially consequential for survival in the wild if released in a stocking program [1]. Body color serves various biological functions, such as camouflage, counter-shading and communication, and the ability to alter coloration playing a crucial role in predation risk, courtship, crypsis, and thermoregulation [6].

The color of the fish skin is generated by absorption, reflection, and scattering of light by the pigments and microstructures within the integument [7,8]. Pigment cells (or chromatophores), derived from the neural crest, are the cellular biological basis of fish body color formation [8,9]. The number of pigment cells, the distribution area, the state of pigment particles in pigment cells, and the reflective ability of reflectors in iridescent cells all affect the body color of fish. Changes in fish body color have two distinct mechanisms: physiological color changes and morphological color changes [10,11]. Physiological color change is mainly caused by short-term stimulation of the external environment and is the more rapid color-change mechanism, consisting of motile chromatophores responses which are caused by the contraction and diffusion of pigment granules in the skin. In contrast, morphological color change is mainly caused by long-term ambient stimulation, resulting from changes in the morphology and density of chromatophores, which is a relative slow process [11,12,13].

The environment can significantly influence the ability of a fish to undergo color changes [8,11]. Among various environmental factors, ambient color stands out as one of the most critical factors influencing chromatophore performance through pigment aggregation or dispersion [8,9,10,11]. Species exhibit specific color-changing patterns in response to different ambient light conditions [14,15,16]. For instance, the southern pygmy perch *Nannoperca australis* tends to display redder coloration in habitats abundant in long wavelengths (550–700 nm) [16]. Studies on sticklebacks *Gasterosteus* spp. reveal that males inhabiting waters dominated by long wavelength light tend to show diminished red coloration [17]. 

In addition to changes in body coloration, fish also exhibit phototactic behavior in response to light stimuli. Research on the snowtrout *Schizothorax waltoni* indicates a preference order for different light colors, with green and blue environments eliciting positive phototaxis responses, while red and yellow environments induce negative phototaxis [18]. Notably, the phototactic responses to light colors may vary within the same species depending on their environmental context. For instance, winter flounder *Pseudopleuronectes americanus* exhibit differences in opsin gene expression before and after metamorphosis. In the larval stage, only RH2 is expressed, indicating high sensitivity to green light, whereas after metamorphosis, SWS2 and LWS cone protein genes are also expressed, suggesting potential changes in spectral sensitivity [19].

Numerous studies have reported the disparities in body coloration between fish reared in traditional high-density intensive culture conditions and their wild counterparts, stemming from variations in their growth environments. Generally, hatchery-reared fish exhibit darker compared to their wild counterparts [20,21,22,23]. For instance, a notable difference in body color was observed between wild and hatchery-reared *Carassius auratus*, with significantly lower lightness and brightness values recorded in the dorsal and abdominal regions of hatchery-reared individuals compared to their wild counterparts [23]. Additionally, there appears to be discernible variance in the visual photosensitive function between hatchery-reared and wild individuals. Notably, research on the second-generation population of hatchery-reared Chinese lenok *Brachymystax tsinlingensis* revealed a significantly higher count of cone cells in the retina, indicative of heightened spectral sensitivity among hatchery-reared individuals [24]. These physiological disparities may influence the perception of environmental light among hatchery-reared individuals, potentially leading to distinct phototaxis behavioral phenotypes. It can be concluded that various factors may contribute to these differences, including genetic selection, ambient light conditions in the rearing environment, and dietary variances [20,23,25,26]. Among these factors, the variance in light environment emerges as a crucial determinant influencing the body coloration of hatchery-reared fish. For instance, Chinese longsnout catfish *Tachysurus dumerili* have a tendency for darker coloration under stronger light intensities [27]. Therefore, it is imperative to elucidate the disparities in body color and environmental color preference among different fish species, including variations within the same species across different environments, to unravel the mechanisms underlying environmental influences on color formation and the adaptive color-changing abilities of fish. However, current research predominantly focuses on the effects of light on fish growth [28], vision [29], and behavior [30,31], with limited research addressing the relationship between light and fish coloration [20,28,32]. Additionally, there is a general lack of research comparing body coloration between hatchery and wild fish, and the influence of brief environmental color exposure on stock fish coloration.

The marbled rockfish *Sebastiscus marmoratus* is a benthic reef species widely distributed in the inshore regions of the East China Sea [33]. Sharing ecological traits with other members of the *Sebastini* tribe, it favors demersal rocky habitats, employing cryptic coloration for camouflage [34,35,36]. In China and Japan, it ranks among the most popular recreational fish species [37]. However, overfishing has precipitated a sharp decline in its wild populations [38,39]. Recent advancements in aquaculture techniques have led to small-scale stocking trials in eastern China, aimed at assessing the viability of supplemental rearing and stocking prior to broader implementation [36,40]. Nonetheless, our prior research has identified notable discrepancies, including notes on body color, between hatchery-reared and wild individuals, potentially compromising the post-release success of cultured fish in natural environments [36]. Given the documented influence of ambient light color on fish physiology and behavior [8,9,10,11,41], elucidating the differences in body color regulation and phototactic responses between wild and hatchery-reared marbled rockfish assumes significant importance, offering insights into the mechanisms governing fish coloration.

This study aimed to compare body coloration, morphological color-changing capabilities, and light color preferences between wild and hatchery-reared marbled rockfish. Additionally, it sought to assess how light color influences the body color performance of hatchery-reared individuals through quantitative analysis of HSB (hue, saturation, and brightness) values of fish skin color. The findings hold potential to illuminate the impact of hatchery rearing conditions on the phenotypic development of marbled rockfish while also informing the design of optimal lighting environments during fish breeding to produce hatchery-reared individuals that closely resemble their wild counterparts, thus enhancing their suitability for release.

## 2. Materials and Methods

### 2.1. Experimental Animals and Holding Conditions

Wild marbled rockfish used in this experiment were captured by angling in July 2021 in the waters surrounding Dongji Island, Zhoushan, China (30°43′ N, 122°46′ E). The sea bottom in this area is predominantly characterized by rocky reef substrate [42]. The marbled rockfish typically inhabit the reef area waters, preferring a depth of approximately 15 m [35]. The monthly average seawater temperature ranges from 8.6 to 27 °C [43]. The average underwater transparency from May to October is 3.1 m [44]. The underwater vegetation in this area primarily consists of warm-temperate species, with a predominance of red and brown algae [43]. The main categories of prey of wild marbled rockfish are fish, amphipods and crabs [45]. Thirty wild marbled rockfish of consistent size (total body length: 10.23 ± 1.50 cm, mean ± SD) were selected and transported to a single circular holding tank (water volume: 1780 L, inside diameter: 1.60 m, height: 0.80 m; equipped with a seawater circulation system) at the Marine Ranching and Fishery Carbon Sink Ecological Function Innovation Laboratory (MRFCSEFIL) of Zhejiang Ocean University. The water temperature in the holding tank was maintained at 20 ± 1 °C, consistent with the prevailing sea temperature during the same period. Continuous oxygenation ensured a dissolved oxygen level of 8–9 mg·L^−1^. The seawater salinity was maintained at 27–28‰, pH levels kept at 8.0–8.3, and unionized ammonia nitrogen levels never reached above 0.1 mg·L^−1^. The photoperiod followed an 11:13 h light/dark cycle, with light intensity ranging from 0 to 500 lx. To mimic natural conditions, enrichment structures in the form of plastic tubes and rocks were introduced into the holding tank. Starting from the third day post-transport, live white prawns *Exopalaemon carinicauda* were provided every other day to ensure a constant supply of live prey.

All the hatchery-reared marbled rockfish were the progeny of wild fish which were caught from the sea around Dongji Island, Zhoushan, China (30°12′ N, 122°40′ E). The hatchery-reared fish used for this study were produced in January 2021 and then reared in indoor nursery ponds (size: 5 m × 6 m × 1 m) in a commercial hatchery (Xixuan Technology Island) in Zhoushan, China, according to the standard methodology for the intensive culture of this species [46]. The water temperature in the nursery pond was maintained between 23 and 25 °C. The stock density was between 5000 and 10,000 individuals per cubic meter. The hatchery-reared fish were fed with commercial pellet feed (Hayashikane Sangyo, Co., Ltd., Yamaguchi City, Yamaguchi Prefecture, Japan; composition: crude protein ≥ 50.0%, lipid ≥ 6.0%, fiber ≤ 3.0%, and ash ≤ 17.0%). In early December 2021, thirty size-matched individuals (total body length: 10.42 ± 1.60 cm, mean ± SD) were selected and transported to a holding tank at MRFCSEFIL of Zhejiang Ocean University. The water temperature in the holding tank was kept at 17 °C, which was consistent with the temperature in the sea during the same period. The dissolved oxygen was ≥6.0 mg/L, unionized ammonia nitrogen was maintained at <0.05 mg/L, salinity was 28‰, and pH was kept at 8.0–8.3. The photoperiod followed an 11 h light/13 h dark cycle, with the light intensity maintained at 200–500lx. From the third day after transport, fish were fed twice daily (at 9:00 and 18:00; to apparent satiation) with commercial dry pellets as mentioned above. 

### 2.2. Experimental Protocol

#### 2.2.1. Disparities in External Body Color between Wild and Hatchery-Reared Marbled Rockfish

Twenty fish from each group (hatchery-reared and wild) were randomly selected for body color determination on the second day following transportation to the laboratory holding tank. The age of individuals was determined from their scales (microscope Olympus BX43; Olympus Life Science, Tokyo, Japan). After removing from water, the experimental fish were swiftly anesthetized using moderate level of tricaine methane sulfonate (MS-222; 100 mg·L^−1^, 30 to 45 s). The anesthetized fish was then put on a white background panel in a controlled studio environment with LED lighting for photographing (40 × 40 × 40 cm). Photographs of the experimental fish were captured using a tripod-mounted digital single-lens reflex camera (Nikon D3400, Nikon, Minato City, Tokyo, Japan) equipped with an 18–55 mm lens (AF-S DX 18–55 mm F/3.5–5.6 G VR II, Nikon, Minato City, Tokyo, Japan) positioned at a standardized distance of 30.0 cm above the fish. Before capturing any images, the camera was calibrated for white balance. Camera settings, including aperture value, exposure time, ISO, and aperture settings, were consistent across all photographs (aperture: f/7.1; exposure: 1/200 sec; ISO: 100; exposure compensation: −1.3). These standardized photographs were utilized for subsequent body color analysis of the experimental fish.

#### 2.2.2. Differences in Morphological Color-Changing Ability between Wild and Hatchery-Reared Marbled Rockfish

The experimental setup consisted of eight interconnected glass tanks (dimensions: 35 × 30 × 30 cm) with a shared recirculating system. Each tank was equipped with a 9W, E27 LED bulb (with power adjustment) suspended above it, and the top of each tank was covered with filter paper of corresponding hues (red [peak at 635 nm], yellow [peak at 581 nm], blue [peak at 453 nm], green [peak at 516 nm]) to induce various ambient light colors. A standardized light intensity of 50 lux was maintained across all tanks, and a 12 h light cycle (12L:12D) was followed. To prevent external ambient light interference, all experimental tanks were enveloped in opaque black film.

After 3 days of recovery from transportation and handling stress, twenty fish from each group (hatchery-reared and wild) were randomly selected for the morphological color-changing ability test. The fish were divided into two categories (‘Hatchery-reared’ or ‘Wild’) × 4 ambient color groups (red, yellow, blue, green), each consisting of 5 experimental fish. These groups were recorded as Wild Red (WR), Wild Yellow (WY), Wild Blue (WB), Wild Green (WG), Hatchery-reared Red (HR), Hatchery-reared Yellow (HY), Hatchery-reared Blue (HB), and Hatchery-reared Green (HG). All experimental fish were kept in the tanks for 10 days and fed to apparent satiation once daily (08:00) with commercial dry pellets. Culture conditions and daily procedures during the experiment remained consistent with the acclimation period. Photographs of all experimental fish in each tank were captured on days 0 (pre-experiment), 5, and 10, using the method described in Section 2.2.1. These photographs were used for body color analysis and comparison of the experimental fish.

#### 2.2.3. Variation in Light Color Preferences between Wild and Hatchery-Marbled Rockfish

Ten fish were randomly selected from both wild and hatchery-reared groups to participate in the light color selection preference test. The testing apparatus consisted of a square tank (60 × 45 × 45 cm) with a mounted real-time monitor camera (Hikvision DS-2CD3T46WDV3-I3, Hangzhou Hikvision Digital Technology Co., Ltd., Hangzhou City, China) equipped with night vision functionality and positioned directly above the tank. The tank was partitioned into four interconnected compartments using foam boards, allowing the test fish to navigate freely at the tank bottom. Over each compartment, four distinct color LED bulbs, with adjustable power, were suspended to create diverse light environments (Figure 1). The light intensity in each compartment was maintained consistently at 50 lux, and the water temperature was regulated to 20 ± 1 °C. To eliminate external ambient light interference, all experimental tanks were enveloped in opaque black film. Each experimental fish was placed in the middle of the testing tank, and then the lights were switched on to start the experiment. For each experiment, an individual fish was introduced into the center of the testing tank to commence the hour-long trial. The experimental protocol followed the work of Xu et al. [18] (Figure 2): the lights were sequenced in the order of red, yellow, green, and blue across the four tank areas, with the duration of time spent in each region recorded through camera video analysis. Following each trial, the light color sequence was systematically rotated clockwise to establish a new testing light environment for subsequent experiments. To assess selection preferences for different light colors and darkness, four light colors were paired with darkness and arranged across the four regions, with the light of the same color placed diagonally. The color arrangement was rotated clockwise after each trial to examine whether the location of the light area influenced the fish’s selection during the experiment. An additional experimental treatment, involving the absence of light (dark treatment), was conducted as a control measure.

#### 2.2.4. Body Color Analyses

To quantify the body color of experimental rockfish, the hue–saturation–brightness (HSB) model [47] was used, following the method described by Yasir and Qin [48]. In this model, H represents hue (°), S represents saturation (%), and B represents brightness (%). Image analysis was conducted using Adobe Photoshop software 2023 (Adobe Inc., Mountain View, CA, USA), employing the rectangle tool to draw the sampling frame. To ensure consistency, all images were zoomed so that the top right corner of the sampling frame aligned with the base at the end of the first dorsal fin, and the bottom right corner was aligned with the base at the beginning of the anal fin (Figure 3). Subsequently, the sampled portions of each image were subjected to pixelated mosaicking, and then evenly divided into six equal parts. A color sampler was used to analyze each part separately, followed by analysis using the color sampler function to obtain the average HSB data.

#### 2.2.5. Calculations

We used the time proportion *F* (in %) to indicate the preference of fish for different light colors:$$F=\frac{f}{N}\times 100$$
where *f* is the time the fish spent in an area with a specific light color; and *N* is the total time of the test.

### 2.3. Statistical Analyses

All data were analyzed in R (R Core Team, Vienna, Austria). The data on body color (hue, saturation, and brightness) of wild and hatchery-reared marbled rockfish were analyzed using a multivariate analysis of variance (MANOVA), followed by individual analyses of variance (ANOVA) for each of the three included variables. Changes in color variables (‘*col.var*’; hue, saturation, or brightness) over ten days, under different ambient light color conditions, were analyzed separately using linear mixed models (R package: lme4 [49]) including the factors of origin (‘*ORIG*’; wild or hatchery), experimental day (‘*DAY*’; 0, 5 or 10), light color (‘*LCOL*’; blue, green, red, or yellow) and their interactions. Fish identity (‘*ID*’) was added as a random factor to account for repeated measures of the same individuals. The full model structure, in lme4-syntax, was as follows: *col.var* ~ *ORIG* * *DAY* * *LCOL* + (1|*ID*). Effects were generally evaluated based on a conservative approach where only non-overlapping 95% confidence intervals for the model-estimated means were considered different (due to low sample size for each factor combination and relatively complex models). In a few cases, where the interval comparisons tended towards particularly interesting (but non-clear) effects, specific pairwise contrasts were applied (Tukey’s method for *p*-value family-wise adjustment for multiple comparisons; R package: emmeans [50]). As ad hoc analyses, separate MANOVAs (as above) were run for each light color, using data for the end result at day 10.

Preference of arena area under dark conditions was evaluated using (i) MANOVA to compare fish from different origins (hatchery-reared vs. wild) and (ii) one sample *t*-tests for each area separately (all fish pooled, irrespective of origin), using a null hypothesis that the average percentage of time spent in each area is 25% (H_0_: μ = 25%). No *p*-value adjustments for multiple comparisons were applied to maintain the sensitivity of the test to detect signs of deviance from the expected 25%.

Preference for ambient light color [i.e., percentage time (‘*perc.time*’) spent in each a light color area (‘*COL*’)] was analyzed as logit-transformed values using a linear mixed model (R-package: lme4); syntax: logit(*perc.time*) ~ *ORIG* * *COL* + (1|*ID*). The model results were evaluated based on analysis of deviance (ANODEV with type III Wald χ^2^ tests; R package: car [51]) and post hoc pair-wise comparisons of contrast estimates (as above).

To analyze preference for colored ambient light vs. dark areas, a linear mixed model was applied where percentage of time in dark areas (‘*perc.time.dark*’) was the dependent variable (logit-transformed). Light condition treatment (‘*LCTR*’; blue/dark, green/dark, red/dark, and yellow/dark; see Figure 2, condition 2) was used as a factor together with origin (as defined above). A model evaluation suggested that the interaction (LCTR * ORIG) had no effect (*p* = 0.556); so, the model was reduced to not include it (hence assuming that both hatchery and wild fish respond similarly to the different light conditions): *perc.time.dark* ~ *LCTR* + *ORIG* + (1|*ID*). As above, the model results were followed by pair-wise comparisons of contrast estimates.

Unless stated otherwise, differences were considered to be significant at *p* < 0.05. Graphics were produced using the ggplot2 [52] and cowplot [53] R packages.

## 3. Results

### 3.1. Body Color

#### 3.1.1. Differences in Body Color between Wild and Hatchery-Reared Marbled Rockfish

Multivariate analysis of differences in body color (hue, saturation, and brightness) between hatchery-reared and wild rockfish showed that the two groups differed (MANOVA: Pillai’s trace = 0.727, *F*_3,35_ = 31.04, *p* < 0.001). Individual ANOVAs for each color variable showed that hue (*F*_1,37_ = 9.79, *p* = 0.003) and saturation (*F*_1,37_ = 91.13, *p* < 0.001) were significantly different (Figure 4A,B). Brightness did not differ significantly (*F*_1,37_ = 1.30, *p* = 0.262; Figure 4C). The hue values of all fish were within a red-magenta (330°) to red (360°) color temperature span, with wild fish on average being slightly closer to red (raw data mean ± SE: 354.28 ± 2.27°) than hatchery-reared conspecifics (343.03 ± 2.81°; Figure 4A). Saturation was substantially higher in wild fish (27.22 ± 2%) than in hatchery-reared fish (7.22 ± 0.4%; Figure 4B).

#### 3.1.2. Changes in Body Color of Marbled Rockfish under Varied Ambient Light Colors

The analyses of hue, saturation and brightness all indicated differences between hatchery-reared and wild rockfish when reared under different ambient light colors for 10 days. Under the conservative approach of only considering non-overlapping confidence intervals as different, differences in hue between hatchery-reared and wild fish at day 0 were not detected (Figure 5A–D), which is a consequence of the conservative nature of the comparisons rather than an absence of differences (day 0 values were analyzed separately in the above test of differences in body color; i.e., with a statistical test, they are indeed different). Regardless of ambient color, wild rockfish did not change their hue values in any clearly discernable way (Figure 5A–D). Hatchery-reared rockfish had clearly decreased hue values after rearing under all ambient light colors (Figure 5A–D), except for day 10 in blue light (Figure 5A). There were no clear differences among different ambient light colors.

Saturation was clearly different between hatchery reared and wild rockfish at days 0 and 5 (Figure 5E–H). Like the results for hue, wild fish did not appear to change their saturation over time (Figure 5E–H). At day 10, the difference between hatchery-reared and wild fish reared under green light was not clear anymore, and similar patterns of hatchery-reared fish approaching the saturation of wild fish were seen in all light color treatments; however, it was not possible to clearly discern the effects over time within the hatchery-reared fish (all confidence intervals were overlapping from day 0 to day 10). Looking specifically at pairwise contrasts between day 5 and day 10 for the hatchery fish reared under green light, the increase in saturation appears significant (difference: 8.8 percentage units; *t* = −2.54, *p* = 0.034); equivalent contrast analyses for the other light color treatments revealed no significant differences; 0.245 > *p* > 0.09). There were no clear differences among different ambient light colors.

For brightness, no clear effects within origin groups could be seen over time (Figure 5I–L); however, under blue light the difference between hatchery-reared and wild fish became apparent at day 5 due to both groups diverging slightly in their mean estimates (but primarily caused by wild fish increasing in brightness; Figure 5I). Similarly, under red light, the two origin groups diverged towards displaying differences in brightness at day 10, here caused by hatchery-reared fish decreasing in brightness (Figure 5K). Under green and yellow light, the brightness was largely similar between origin groups.

Overall, ad hoc MANOVAs on color variables at day 10 revealed that significant differences between hatchery-reared and wild fish were still present under all ambient light colors (*P*_blue_ < 0.001; *P*_green_ = 0.036; *P*_red_ = 0.002; *P*_yellow_ < 0.001).

### 3.2. Light Color Preference

Hatchery-reared and wild rockfish exhibited similar average distribution patterns over different arena areas under no-light conditions (MANOVA: Pillai’s trace = 0.106, *F*_1,18_ = 0.444, *p* = 0.775). No significant deviations from the expected 25% of time spent in each area were detected (Table 1); however, at the individual level, some fish spent disproportionate amounts of time in certain areas. Hence, while distribution patterns were not always random, there were no systematic preferences detected.

#### 3.2.1. Light Color Preferences

Analysis of deviance indicated significant interaction effects between light color and origin (COL: *χ*^2^ = 6.54, *p* = 0.088; ORIG: *χ*^2^ = 6.54, *p* = 0.054; ORIG * COL: *χ*^2^ = 6.54, *p* = 0.033). Pairwise contrasts of origin effects in each light color area indicated that hatchery-reared and wild fish differed in the time spent in the red ambient color area, with wild fish spending significantly more time in this area (*t* = −2.04, *p* = 0.045; Figure 6). A trend was detected for time spent in the blue area, but it was not significant (*t* = 1.93, *p* = 0.058). For green and yellow areas, no tendencies for differences were found (green: *p* = 0.624; yellow: *p* = 0.298). Pairwise contrasts among color areas within origin groups revealed significant preference for red areas over any other color in wild fish (all *p* < 0.001). No differences among blue, green, or yellow areas were found for wild fish (all *p* > 0.850). No significant color preference was detected for hatchery-reared fish (all contrasts with *p* > 0.066); the largest difference was found between red and green areas. It is worth noting that the majority of the hatchery-reared fish spent more than 25% in the red area, indicating a possibility for a slight preference for red, which might be detectable with larger sample sizes than *n* = 10 (i.e., preference for red should not be discarded based on these results, even though it was not strictly verified).

#### 3.2.2. Selection Preferences of Wild and Hatchery-Reared Marbled Rockfish under Varied Ambient Light Colors and Darkness

Given the model being reduced to not include the interaction between light color and origin of the fish (which was non-significant in the initial full model: *χ*^2^ = 2.066, *p* = 0.556), the results are restricted to detecting general differences between light color treatment and general differences between hatchery-reared and wild fish. With respect to light color treatment, the light color being paired with dark areas had a significant effect on the preference for the dark area (*χ*^2^ = 14.79, *p* = 0.002), but there was no effect of origin (*χ*^2^ = 0.604, *p* = 0.437). Hence, the paired contrasts focused on the main effect of treatment (fish from different origins pooled). The fish showed a stronger preference for the dark environment when paired with blue-light areas than when paired with green- or red-light areas (blue vs. green: *t* = 2.82, *p* = 0.032; blue vs. red: *t* = 3.36, *p* = 0.007) (Figure 7). A trend was noted for the comparison of yellow- vs. red-light treatment (*t* = −2.38, *p* = 0.092) and the other contrasts were clearly non-significant (all *p* > 0.26). Inspecting the 95% confidence intervals (Figure 7), it is apparent that the fish do not prefer dark areas over red- or green-light areas. It is also clear that dark areas are generally preferred over blue-light areas. The same is likely the case when paired with yellow-light areas, but the preference for dark appears slightly lower in this case, and for hatchery fish, the confidence interval slightly overlaps with the 50% line (Figure 7), making the conclusion uncertain. 

## 4. Discussions

### 4.1. Body Color Differences

The rearing environment significantly impacts the body coloration of hatchery-reared marbled rockfish, leading to observable differences compared to their wild counterparts. Similar effects have been seen in other species. For instance, hatchery-reared clownfish *Amphiprion ocellaris* exhibit less vibrant body colors than their wild counterparts, and upon transfer of wild conspecifics to indoor culture, their body color transitions from yellow-orange to orange-pink due to changes in carotenoid composition within the epidermis [54]. Similarly, after being captured from the wild and reared in an indoor culture, red porgy *Pagrus pagrus* individuals undergo a darkening of body coloration from silvery red to dark gray, particularly noticeable in the tail and fins [55]. These examples collectively indicate that hatchery rearing can influence fish body color, and that it generally leads to darker and less colorful phenotypes. In our study, the hue and saturation values of body color in wild marbled rockfish were significantly higher than those of hatchery-reared individuals, while no significant difference was observed in brightness values. Hence, wild marbled rockfish exhibit a more colorful (redder) body compared to hatchery-reared fish, aligning with findings in many of the investigated species mentioned above. Fish skin color is a complex trait determined by a combination of genetic, cellular, physiological, and environmental factors [8,9,10,11,56,57]. Genetic polymorphisms controlled by specific genes contribute to skin color variations in fish, with numerous studies highlighting genetic influences on phenotypic differences [58]. Additionally, fish can enhance skin and flesh coloring by ingesting foods containing natural pigments, such as astaxanthin [59,60,61]. Environmental factors can also induce the transfer of melanosomes within the pigment cells of teleost, resulting in changes in skin color or hue. These factors collectively influence the overall body color of fish [62,63]. Given that all hatchery-reared fish in our study were progeny of wild-caught parents, it is likely that the genetics of the cultured experimental fish remain unchanged relative to those of the wild-caught fish. Therefore, the observed phenotypic divergence in body color among hatchery fish is likely attributed to rearing conditions and/or dietary differences [64,65,66]. While the potential for higher mortality rates in wild conditions compared to hatcheries could influence genetic patterns between wild and hatchery fish, this aspect is not the focus of our study. Studies have demonstrated that fish can morphologically alter their body color by adjusting the number and size of melanophores during long-term acclimation to light or dark environments [67,68]. In unpublished results from another of our studies, adding astaxanthin to the feed effectively improved the vividness of body color in hatchery-reared marbled rockfish (authors’ unpublished results). Hence, we hypothesize that environmental factors and feeding practices contribute to the observed body color differences between wild and hatchery-reared marbled rockfish. Since fish body color serves various biological functions in the wild, these conspicuous disparities between wild and hatchery-reared marbled rockfish may have negative consequences for the post-release performance of hatchery-reared individuals in natural environments. Therefore, it is crucial to make modifications to the hatchery environment and operation to minimize these divergences in body color phenotype.

### 4.2. Effects of Environmental Ambient Light Color on Body Color

As adaptable organisms, fish possess the remarkable ability to adjust their body color to match their surroundings. Changes in environmental light color and/or intensity may profoundly influence the formation of body pigmentation, ultimately affecting overall body coloration. Large yellow croakers *Larimichthys crocea* display a rapid responsiveness to environmental light in body coloration between day (silver-white) and night (golden yellow) [69]. In the present study, wild marbled rockfish maintained highly consistent body coloration following a ten-day exposure to various light colors, whereas hatchery-reared marbled rockfish exhibited a noticeable change under some ambient light colors. This indicates that wild and hatchery-reared rockfish have different morphological color responses to changing ambient light, a response that is reasonable given that they start the exposure treatment with different coloration. Given that significant changes are only seen in hatchery-reared fish, their coloration appears less stable (or more flexible) to environmental influence; whether this is a good feature or not is not ascertained from our studies. We hypothesize that color changes observed in hatchery-reared fish may stem from more flexible aggregation or dispersion of pigment cells as compared to wild fish. Long-term acclimation to specific light environments has been shown to alter the size and density of chromatophores, facilitating enhanced environmental adaptation [15,70,71]. Further investigation is needed to elucidate the precise impact of light color on pigmentation and its underlying regulatory mechanisms. 

From a stocking perspective, it is worth noting that the color adaptation of hatchery-reared fish did entirely move toward a wild-individual coloration (i.e., saturation tended toward wild-individual values, but hue diverged further). While some changes were rapid, a longer time is likely needed for stronger responses, and other factors might have to be added to reach a wild-like coloration, such as high carotenoid diet and/or physical enrichment structures. Given the apparent flexibility in coloration seen in hatchery-reared fish, we hypothesize that individuals stocked into the wild will eventually attain a wild-like coloration. Nevertheless, given that hatchery fish may be most vulnerable just after release, it is potentially beneficial to pre-adapt their coloration before the release.

### 4.3. Selection Preferences for Ambient Light

Fish can distinguish between different light colors and develop color preferences [72]. Various light wavelengths can induce changes in the photoreceptors of fish, subsequently triggering locomotor activity and influencing the fish’s movement towards or away from the light source [73]. In this study, both wild marbled rockfish exhibited a marked preference for red ambient light, and a similar pattern was seen for the hatchery-reared individuals (although weaker and not statistically significant). Many fishes possess visual pigment cells adapted to the wavelengths dominating their specific environments and visual sensitivity is heightened when fish are in an ambient light environment that maximizes photon capture [74,75,76]. We hypothesize that the red phototropism observed in marbled rockfish may be attributed to the matching of visual pigments to long wavelengths. According to the spectral theory, most of the red light is filtered out when the sunlight passes through the water layer due to the absorption and reflection of water. Wild marbled rockfish tend to inhabit underwater reef areas with weak light, and their visual characteristics are compatible with the natural habitat environment. Therefore, there is no difference in the choice of the red-black light color combination. Histophysiology studies of the wild marbled rockfish retina indicate that the cones’ light-sensing function is adapted for low-light vision [77]. More detailed studies investigating the differences in photoreceptor function of the retina between wild and hatchery-reared marbled rockfish are necessary to provide more information about marbled rockfish light sensitivity.

## 5. Conclusions

In conclusion, the results of this study revealed significant differences in body color between wild and hatchery-reared marbled rockfish. The hue (H) and saturation (S) values of wild marbled rockfish were significantly higher, indicating a more intense and vibrant body coloration. We hypothesize that environmental factors (and food carotenoid content, as investigated elsewhere) play a pivotal role in the body color expression. Light color preference tests demonstrated that marbled rockfish exhibited a preference for ambient red light, albeit with a more pronounced preference in the wild population compared to their hatchery-reared counterparts. Both wild and hatchery-reared marbled rockfish displayed notable negative phototaxis in yellow and blue ambient light, but not in red or green ambient light. Here, no major effect from hatchery rearing was detected. Notably, our study demonstrates that a mere ten-day rearing period under specific light color ambient conditions can result in changed coloration in hatchery-reared marbled rockfish, but it is not enough to achieve a completely wild-like body color. These insights offer valuable information for guiding modifications to the hatchery environment aimed at producing marbled rockfish with a wild-like phenotype.

## Figures and Tables

**Figure 1 animals-14-01701-f001:**
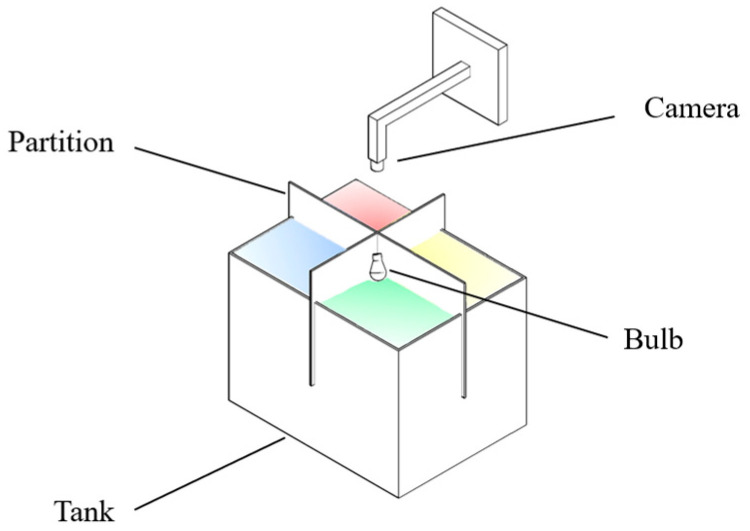
Diagram of the experimental setup.

**Figure 2 animals-14-01701-f002:**
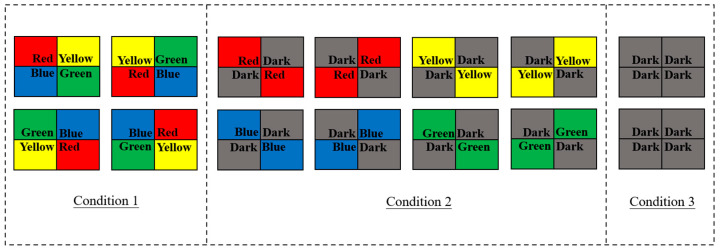
The protocol of light color condition.

**Figure 3 animals-14-01701-f003:**
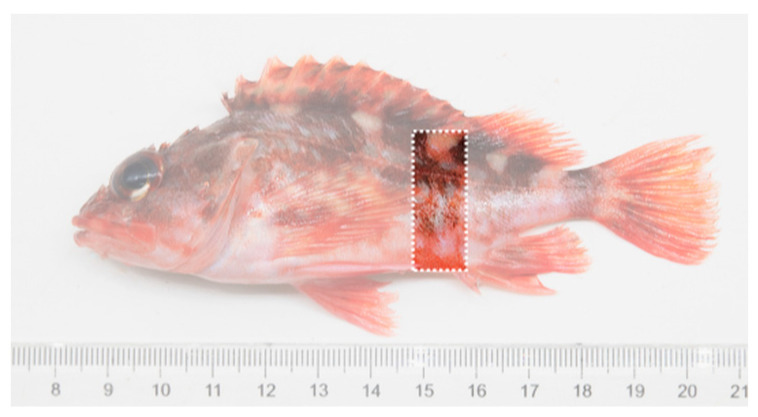
Schematic diagram of the body color sampling area.

**Figure 4 animals-14-01701-f004:**
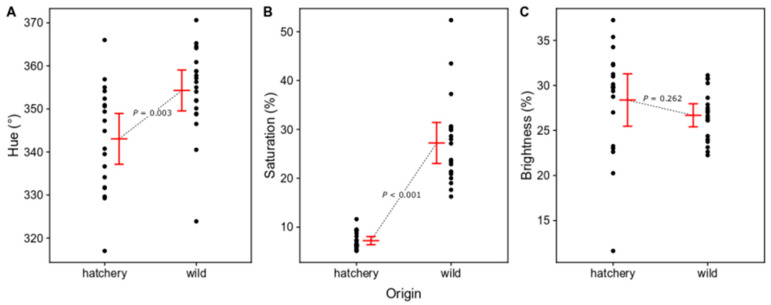
Coloration comparisons of hatchery-reared and wild marbled rockfish. (**A**) Hue values, (**B**) saturation values, and (**C**) brightness values. Mean estimates with 95% confidence intervals are presented in red color and raw data are presented as black dots. *p*-values are derived from separate ANOVA models, following a MANOVA (Pillai’s trace = 0.727; *p* < 0.001).

**Figure 5 animals-14-01701-f005:**
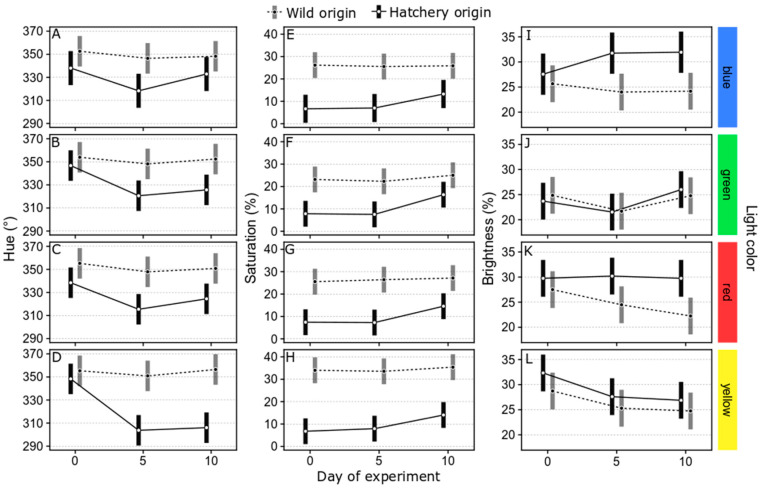
Changes in hue (**A**–**D**), saturation (**E**–**H**) and brightness (**I**–**L**) values of wild and hatchery-reared marbled rockfish reared under different environmental light colors (blue, green, red, and yellow) over ten days. Data are presented based on estimates from linear mixed models for each color variable (mean estimate with 95% confidence interval).

**Figure 6 animals-14-01701-f006:**
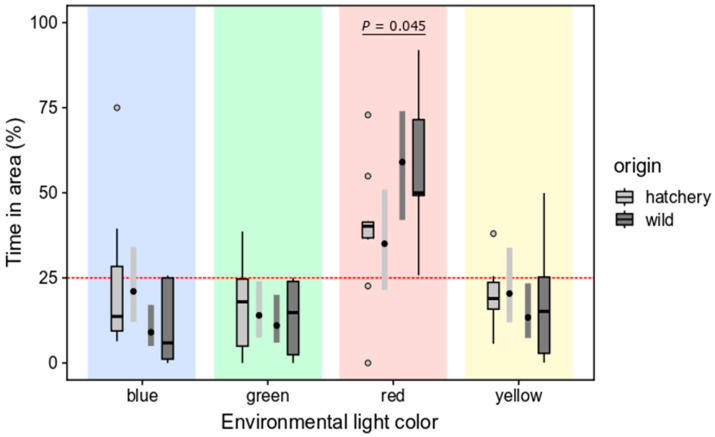
Time spent (as %) in each of four areas with different environmental light color, for wild and hatchery-reared marbled rockfish. Data are presented (back-transformed from the logit scale) as both Tukey boxplots (with interquartile ranges and outliers depicted) and model estimates (estimated mean and 95% confidence interval). Reported *p*-values indicate significant differences between wild and hatchery-reared fish; if no *p*-value is reported, then no significant differences were detected). For wild fish, the red-light environment was significantly preferred over all other light colors (all contrasts with *p* < 0.001); for hatchery-reared fish, no light environments were significantly different (all *p* ≥ 0.067). Red line at y = 25% indicates the expected time spent in an area in the absence of any light color preference.

**Figure 7 animals-14-01701-f007:**
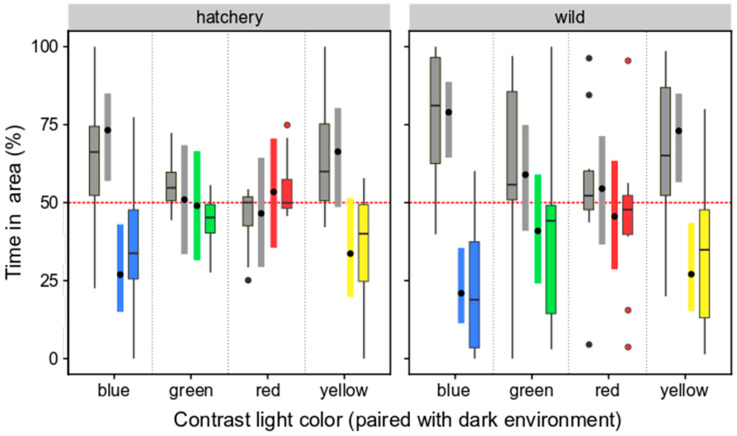
Comparisons of time spent (%) in dark or colored-light environments for hatchery-reared and wild marbled rockfish. Data are presented for both dark and colored light, as both Tukey boxplots (with interquartile ranges and outliers depicted) and estimates from the hypothesis test model (linear mixed model; estimated means, with 95% confidence interval). Red line at y = 50% indicates the expected time spent in an area in the absence of any light color preference.

**Table 1 animals-14-01701-t001:** Distribution of marbled rockfish in different areas (see Figure 1) in complete dark conditions (Figure 2, condition 3). Data are presented as percent of time spent in each quadrant (Area 1–4), with the 95% confidence interval for the mean within parentheses. Reported *p*-values refer to single-sample *t*-tests for each area (alternative hypothesis: true mean not equal to 25%).

	Area 1 (*F*_1_)	Area 2 (*F*_2_)	Area 3 (*F*_3_)	Area 4 (*F*_4_)
Average time (%)	17.4 (8.1–26.7)	34.9 (18.7–51.1)	22.2 (11.6–32.7)	25.6 (17.2–34.0)
*p* (H_0_: μ = 25%)	0.102	0.215	0.579	0.884

## Data Availability

The original contributions presented in the study are included in the article material. Further inquiries can be directed to the corresponding author.

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
