# Peer review of "Disparities in Body Color Adaptability and Ambient Light Color Preference between Wild and Hatchery-Reared Marbled Rockfish (Sebastiscus marmoratus)"

_animals, 2024, doi:10.3390/ani14111701_

Round 1
Reviewer 1 Report
Comments and Suggestions for Authors
The manuscript contains quite a lot of interesting and important information, but I have many comments, questions, and suggestions for its improvement. I believe that it can be published if the authors will be able to address the comments below adequately:
General Comment:
The study employs various methods to examine the coloration of marbled rockfish, but these methods appear disconnected and could potentially be presented as separate studies. The distinct parts of the manuscript lack cohesive integration, which fragments the overall narrative. I recommend revising the Introduction and Discussion sections to link these segments better, thereby enhancing the manuscript's coherence and unity.
Materials and Methods:
A comprehensive description of the environments where wild fish were captured—including depth, bottom type and color, underwater vegetation, and the presence of prey and predators—is essential for a thorough understanding of the observed differences between wild and hatchery fish. Similarly, details on the conditions under which hatchery fish were maintained, such as depth, temperature, wind exposure, fish density, and diet, are crucial.
The maturation status and age of the fish should be described, as behavior toward coloration might vary significantly between adult and juvenile stages. It would be helpful to clarify whether hatchery and wild fish were of similar ages and maturity status.
The manuscript does not explain why different foods were provided to wild and hatchery fish during the study, given that diet is known to affect body coloration. This discrepancy could significantly influence the study's outcomes.
The interval between removing the fish from water and photographing them remains unclear. This is an important detail because fish can change their coloration rapidly.
The choice of using darkness instead of white light as a control condition is questionable. White light, symbolizing neutrality, might have been a more suitable control, as darkness could cause fish to hide, potentially skewing results.
The selection of a specific body part to measure color raises two important questions. First, how was the standardization of the orthogonal plane where the color was measured approached? Fish, even those similar in size, can vary somewhat in size and shape (these differences may arise, for instance, between hatchery and wild fish), yet the size of the frame used for all fish is the same, and the resolution of the pictures is also the same (as assumed, though not explicitly mentioned in the text). This implies that while the frame dimensions remain constant across fish, the fish themselves may vary. How was this issue resolved?
Second, the body part used for color analysis (the tested area) includes dark stripes on a lighter background. As the size of these stripes is comparable to the tested area, their size and presence may significantly affect the results. From experience, such stripes can be quite variable. The fact that the authors did not address this consideration complicates the interpretation of the results. The observed differences could be due to uniform changes in both dark and light parts of the tested area, unequal changes in light and dark parts, or a varying ratio of dark stripe areas within the tested region. How were these potential sources of variability accounted for in the analysis? The absence of a standardized color scale is also troubling, as it would facilitate more precise quantification of color metrics and comparison with other studies. This omission potentially diminishes the study's impact.
It is not clear why a standard color scale is not used to allow more exact quantification of color metrics and easier comparison with results from other studies (made by other researchers as well as the authors of this research). From my point of view, this reduces the value of this study.
I do not quite understand the concept of non-overlapping for quantifying differences between samples. In fact, even if distributions do not overlap, especially if the sample size is small, it does not guarantee that they belong to different populations. For instance, the results of such a comparison may greatly depend on frequency distribution patterns. I believe that just standard approaches, i.e. providing exact p-values, are more informative.
Results:
It would be beneficial to know if there was any correlation studied between different color metrics, such as Hue and Saturation. Given their similar patterns in wild versus hatchery fish, understanding their correlation is vital for accurate interpretation. I recommend employing principal component analysis across individuals to better understand relationship between all three color metrics.
Line 396: The term "swimming" seems misused here. Consider rephrasing.
Line 407: The p-value equaling to 0.045 indicates marginal significance, especially if multiple comparisons were not conducted. Was there any correction applied for multiple testing?
Discussion:
The manuscript suggests that color phenotypic differences between wild and hatchery fish are caused by differences in environmental conditions because the hatchery fish are progeny of the wild fish caught in the same locality. However, genetic differences between wild and hatchery fish are also possible due to potential mortality in wild conditions which is usually high and not random, while in the hatcheries the selection is much weaker due to low mortality and may have a different pattern. This also needs to be taken into account while interpreting the results. By the way, where did the release of hatchery fish into the wild take place? Is it possible that some of the wild fish used in the experiment are actually of hatchery origin?
Reviewer 2 Report
Comments and Suggestions for Authors
Please see attachment

Extensive editing of English language required
Round 2
Reviewer 1 Report
Comments and Suggestions for Authors
Reviewer 2 Report
Comments and Suggestions for Authors
The author has made revisions according to the reviewer's comments and I agreed to accept the current format
Author Response
There is no new comment from reviewer 2. We would like to express our thanks for reviewer 2 once again for the previous valuable feedback, which has undoubtedly improved the quality of our manuscript.
Best regards.
Yours sincerely,
Yulu Qi